# Vancomycin-Resistant *Streptococcus thoraltensis*: A Case Report of Bacterial Endocarditis and Review of Literature on Infections Caused by This Pathogen

**DOI:** 10.3390/microorganisms12030566

**Published:** 2024-03-12

**Authors:** Roxana Mihaela Chiorescu, Sandor Botond Buksa, Alexandru Botan, Mihaela Mocan, Carmen Costache, Dan Alexandru Toc

**Affiliations:** 1Department of Internal Medicine, “Iuliu Hațieganu” University of Medicine and Pharmacy, 400012 Cluj-Napoca, Romania; roxana.chiorescu11@gmail.com (R.M.C.);; 21st Medical Clinic, Internal Medicine Department, Emergency Clinical County Hospital, 400006 Cluj-Napoca, Romania; 3Emergency Clinical County Hospital, 400000 Cluj-Napoca, Romania; 4Faculty of Medicine, “Iuliu Hațieganu” University of Medicine and Pharmacy, 400012 Cluj-Napoca, Romania; 5Department of Microbiology, “Iuliu Hațieganu” University of Medicine and Pharmacy, 400012 Cluj-Napoca, Romania

**Keywords:** bacterial endocarditis, *Streptococcus thoraltensis*, large vegetation

## Abstract

*Streptococcus thoraltensis* is a rare species, part of the viridans streptococcus group, found initially in rabbits and pigs, which can be vancomycin-resistant. We present the case of a 65-year-old patient, a smoker and chronic alcohol consumer with chronic obstructive pulmonary disease (COPD) and multiple dental foci who had been diagnosed with bacterial endocarditis caused by *Streptococcus thoraltensis*. The particular elements of the case consisted of an atypical clinical presentation with diarrheal stools, abdominal pain, concomitant damage to the aortic and tricuspid valves, the presence of large vegetations (>2 cm), and a vancomycin-resistant strain of *Streptococcus thoraltensis*. The evolution of the patient was unfavorable due to septic embolisms, respiratory failure requiring orotracheal intubation, and septic and cardiogenic shock. Infections with *Streptococcus thoraltensis* are challenging to treat because of the severity of the clinical form it causes and the pattern of antibiotic resistance in this germ. Based on our brief review, *Streptococcus thoraltensis* is an extremely rare human pathogen previously described as the etiologic agent of infectious endocarditis in only one case.

## 1. Introduction

*Streptococcus thoraltensis* is a bacterium that was described by Devriese et al. in 1997 [1]. It was initially isolated from the vaginal fluids of sows. However, in 2010, Boro et al. isolated *Streptococcus thoraltensis* from rabbit feces, suggesting for the first time that this bacterium may present a wider distribution than previously thought [2].

*Streptococcus thoraltensis* is an alpha-hemolytic streptococcus, part of the viridans group. Usually, it produces small and non-pigmented colonies. It is catalase-negative, oxidase-negative, and non-motile [1]. In addition, it has a specific biochemical profile with positive reactions for arginine hydrolysis, production of beta-glucuronidase, and production of acid from various sources like ribose, mannitol, and inulin [1]. These characteristics make this species difficult to diagnose using only conventional methods.

The involvement of *Streptococcus thoraltensis* in human infections remains unclear. In 2017, AlWakeel et al. described the isolation of this bacterium as part of the normal nasal and oropharyngeal flora in fuel workers [3]. Exposure to toxic fumes may change the content of the normal flora, posing a positive influence on the proliferation of *Streptococcus thoraltensis*. Al-Tamimi et al. described the isolation of a vancomycin-resistant strain of *Streptococcus thoraltensis* from the nasal cavity of a healthy person [4]. This finding provides an interesting insight into the pathogenetic potential of this bacterium considering that the isolated strain was *vanA*-negative. Future infections with a resistant strain may pose some difficulties regarding the proper treatment. 

The aim of this paper is to describe a case of infective endocarditis produced by a vancomycin-resistant strain of *Streptococcus thoraltensis* and to review the human infections reported by this elusive bacterium.

## 2. Case Report

We present a case of a 65-year-old male patient who addressed the Emergency Department (ED) of the Cluj County Emergency Hospital for the following symptoms: diarrhea, flatulence, and extensive edema in the lower limbs. The patient had no significant family history. He had been previously diagnosed with toxic dilated cardiomyopathy, recurrent atrial fibrillation (AF), congestive heart failure, chronic obstructive pulmonary disease (COPD), and toxic liver cirrhosis due to chronic alcohol consumption. The patient in question was a chronic smoker, having consumed an average of 20 packs of cigarettes annually for a period of 30 years. Additionally, the patient had a history of long-term alcohol consumption, with daily intake exceeding 40 g for the past 40 years. In terms of medication, the patient was receiving treatment for a variety of conditions, including spironolactone and furosemide for heart failure, ramipril for hypertension, amiodarone for arrhythmia, and tiotropium inhalers for chronic obstructive pulmonary disease (COPD). The clinical presentation commenced subtly approximately two months prior, characterized by bilateral lower limb edema, intermittent palpitations, asthenia, and fatigue. A syncopal episode occurred six weeks before the ED presentation. The patient was hospitalized, and following ECG Holter monitoring, the diagnosis of repetitive paroxysmal AF was established. Echocardiography revealed the appearance of dilated cardiomyopathy without visualizing transthoracic valvular vegetation at the level of valves. At the same time, the patient was diagnosed with toxic liver cirrhosis Child A and BOPC stage II GOLD. The above treatment had been initiated. The patient did not come for an outpatient check-up.

After admission, the patient reported experiencing more than three episodes of watery diarrhea per day, devoid of pathological constituents, accompanied by a weight loss of four kilograms over the previously mentioned period. Upon physical examination in the Emergency Department (ED), findings included obesity with a body mass index (BMI) of 31 kg/m^2^, a body temperature of 37.5 °C, and a state of obnubilation. The clinical examination revealed the following findings:Arterial blood pressure: 140/70 mmHg; heart rate: 100 beats per minute.Rhythmic heart sounds, grade III diastolic mitral murmur heard at the mitral focus, grade II systolic murmur also heard at the mitral focus, stressed second cardiac sound, holosystolic murmur detected in the right parasternal space.Distended abdomen, mobile with breathing, painful on deep palpation; liver located 2 cm below the costal rim; and splenomegaly.Presence of chronic periodontitis, characterized by multiple root remnants.Dehydrated skin and mucosa, itchy and disseminated spots with cafe au lait appearance, edema of the lower limbs, clubbing fingers, hypotonic muscular system.

The etiology of the observed edematous syndrome encompasses potential pathologies such as heart failure, decompensated liver cirrhosis, hypoproteinemia secondary to malabsorption, or nephrotic syndrome. To delineate the precise diagnosis, a battery of diagnostic tests was recommended, including routine laboratory analyses, arterial Astrup gasometry (particularly relevant in cases of respiratory failure), 24 h proteinuria collection, electrocardiography (EKG), echocardiography, abdominal ultrasound, stool examination, and parasitological evaluation of stool samples. The arterial blood gas analysis revealed findings consistent with metabolic acidosis and mild hypoxemia alongside hypocapnia, characterized by a pH of 7.30, a partial pressure of carbon dioxide (PCO_2_) of 30.3 mmHg, a bicarbonate (HCO_3_) concentration of 19.9 mmol/L, and a partial pressure of oxygen (pO_2_) of 70.7 mmHg. The laboratory analyses revealed several significant findings:Presence of an inflammatory syndrome, evidenced by elevated levels of C-reactive protein (CRP) at 9.72 mg/dL, an erythrocyte sedimentation rate (ESR) of 27 mm/h, leukocytosis of 10.23 × 10^9^/L with 90.7% neutrophils (within the range of 30–75%), and an elevated ferritin level of 627 ng/mL.Normochromic normocytic anemia, indicated by a hemoglobin (Hb) level of 10.6 g/dL.Thrombocytopenia, characterized by a platelet count of 35 × 10^9^/L.Evidence of hepatic impairment, including hypoalbuminemia with an albumin level of 2.61 g/dL, decreased cholinesterase activity at 1349 IU/L, and an elevated international normalized ratio (INR) of 1.56.Intrahepatic cytolysis, demonstrated by elevated levels of both direct bilirubin (Bd) at 2.72 mg/dL and total bilirubin (Bt) at 5.3 mg/dL.Renal dysfunction classified as stage 3b according to KDIGO criteria, with elevated serum creatinine levels of 2.22 mg/dL, a calculated creatinine clearance of 28 mL/min, hyperkalemia with a potassium (K) concentration of 5.8 mmol/L, hyponatremia with a sodium (Na) concentration of 126 mEq/L, and hyperuricemia with a uric acid level of 18.10 mg/dL.Elevated levels of N-terminal pro-B-type natriuretic peptide (NT-proBNP) at 1747 pg/dL, suggesting cardiac dysfunction.Abnormal urine analysis revealing leukocyturia, hematuria, and significant proteinuria at 2 g/24 h.Serological testing indicated positivity for hepatitis B surface antigen (HBsAg), negativity for anti-hepatitis C virus (HCV) antibodies, and negativity for human immunodeficiency virus (HIV) antibodies.GDH and toxin A, B tests for *Clostridioides difficile* were negative.

EKG (made at the patient’s presentation): sinus rhythm, HR = 100 beats per minute, QRS electrical axis at 0°, QS appearance in V1–V4, microvoltage without any terminal phase change (Figure 1).

**Doppler echocardiography** showed a non-dilated left ventricle (LV—55/47 mm), grade III aortic insufficiency with a jet that hit the anterior mitral valve, mobile hyperechoic mass located on the aortic valve (3.5/0.95 cm) and on the tricuspid valve (2.5/1.4 cm), grade II tricuspid insufficiency, moderate secondary pulmonary hypertension (PAPs—50 mmHg), a slightly dilated right ventricle (40 mm sub tricuspid), TAPSE of 22 mm, EF of 50%, and free pericardium (Figure 2 and Figure 3).

**A transesophageal ultrasound would have been useful for** better visualization of cardiac structures but was not performed because the patient’s clinical condition did not allow for the examination (obnubilated patient with respiratory failure).

The mobile hyperechoic formations at the level of the aortic and mitral valves primarily raise the suspicion of bacterial endocarditis. The differential diagnosis of these formations is made with thrombi, tumors, or intracardiac structures. 

The abdominal ultrasound exhibited features consistent with cirrhotic liver morphology, including an irregular hepatic contour, dilated portal vein, and a small volume of subhepatic ascites. Additionally, there was noted dilation of the inferior vena cava with decreased respiratory collapse, thickening of the gallbladder walls, splenomegaly measuring 140/50 mm, observable peristalsis in bowel loops, and an absence of pleural effusions. To confirm bacterial endocarditis, three sets of blood cultures were collected at an interval of one hour. Antimicrobial treatment with vancomycin and ceftriaxone was empirically initiated and adapted to creatinine clearance.

All three specimens underwent incubation in the laboratory utilizing the TDR Automated Blood Culture System, resulting in growth. Subsequent cultivation on sheep blood agar yielded alpha-hemolytic bacteria. Gram staining revealed the presence of Gram-positive cocci arranged in chains through a negative catalase test. Further bacterial identification and antimicrobial susceptibility testing were conducted using the Vitek 2 Compact system.

The conclusive diagnosis of infective endocarditis was established, attributed to vancomycin-resistant *Streptococcus thoraltensis*, due to meeting two major criteria outlined for bacterial endocarditis according to the European Infectious Endocarditis Management guidelines (2023).

A series of laboratory analyses, examinations, and specialized consultations were undertaken to delineate the sources of infection and determine the initial site of bacterial endocarditis. These included urine culture (yielding negative results), pharyngeal swabs (yielding negative results), stool sample analysis (yielding negative results), abdominal ultrasound (indicating an absence of acute cholecystitis or intra-abdominal abscesses), and consultation with oro-maxillofacial surgery experts. The consultation revealed findings of poor oral hygiene and subtotal non-prosthetic maxillary ventilation. Recommendations included orthopantomography and dental focus sanitation, scheduled for implementation following clinical and hemodynamic stabilization.

To evaluate the potential presence of septic emboli, a native cranial CT scan was performed, which did not reveal any evidence of hemorrhage or cerebral infarction. The neurological consultation identified the patient as clinically confused with paraparesis and bilateral mutism, albeit without discernible focal neurological deficits. Given the clinical suspicion, a brain MRI was recommended to definitively rule out septic embolism. Additionally, abdominal ultrasound was utilized to rule out splenic or renal infections.

The antimicrobial susceptibility testing results indicated resistance to vancomycin, tetracycline, tigecycline, clindamycin, and rifampicin, with the treatment regimen thus adjusted to ampicillin and gentamicin.

Considering the presence of large vegetations (>2 cm) associated with an increased risk of embolization, consultation with cardiovascular surgery specialists was sought. The recommendation was made to postpone surgical intervention because of too high a surgical risk and to continue antibiotic therapy in accordance with the antibiogram.

Unfortunately, the patient’s clinical course deteriorated, marked by the development of septic emboli leading to coma, respiratory failure requiring orotracheal intubation, and subsequent progression to septic and cardiogenic shock, subsequently resulting in the patient’s death.

## 3. Discussion

Infectious endocarditis (IE) is a septic disease of the endocardium caused by a microbial infection with bacteria or fungi [5].

Risk factors for IE are age over 60 years, the presence of dental foci of infection, rheumatic valvular diseases, valve prostheses, a history of infectious endocarditis, intracardiac devices, intravenous drug use, diabetes, immunocompromised status, and hemodialysis [6]. 

As predisposing factors, we identified the presence of multiple dental foci, age, and immunocompromised status (patient with liver cirrhosis and COPD) in our patient.

The epidemiological profile of IE has changed substantially in recent years, especially in industrialized countries, specifically in terms of modifying predisposing heart lesions. IE now affects more elderly patients, those who have undergone interventional procedures; we find it in wearers of prostheses or intracardiac devices. In the past, it affected young adults more frequently, and the predisposing valvular lesion was rheumatic valvopathy. Change in the spectrum of predisposing lesions was associated with a change in the bacteriological profile of IE. Thus, if oral streptococci were the most common cause of IE in the past, this position is now occupied by *Staphylococcus aureus* [7].

The classification of bacterial endocarditis is based on clinical evolutionary criteria (acute and subacute IE) depending on the location of the infection (on native valves, prosthetic, intracardiac devices or right heart), the way of acquiring the infection (community or nosocomial infection, IE in the patient on hemodialysis, IE in the patient with orotracheal intubation, IE in intravenous drug users), and the etiological agent [5,6]. 

The *Streptococcus viridans* group remains the main group of microorganisms that cause subacute IE. *Staphylococcus aureus* is the main microorganism that causes right heart IE found in intravenous drug users. *Enterococcus faecalis* is an etiological agent of IE in neoplastic patients and those in whom endoscopic maneuvers have been performed on the gastrointestinal tract [8].

The insidious clinical onset led us to subacute endocarditis, and the presence of undocumented splenomegaly on an ultrasound performed two months earlier certified this type of endocarditis. The peculiarity of the case consisted in the fact that the etiological agent was represented by *Streptococcus thoraltensis*, which also proved to have exceptional resistance to antibiotics.

From a clinical point of view, the essential elements for diagnosis are fever and new heart murmur or a change to the character of pre-existing murmurs [5].

In addition to the characteristic clinical signs (fever, new murmur), other signs should raise suspicion of bacterial endocarditis (general symptoms such as unexplained malaise, weakness, arthralgia, and weight loss; embolic events of unknown origin in the brain, spleen, or kidneys; atrioventricular conduction disorders; palming; skin lesions such as Osler or Janeway nodules; and ophthalmologic manifestations (Roth spots)) [9,10,11].

Elderly and immunocompromised patients, as well as those with endocarditis of the right heart, may present with atypical signs of endocarditis, as was the case with our patient. A high degree of clinical suspicion is required to diagnose these patients.

Key aspects of infective endocarditis (IE) diagnosis rely on a combination of echocardiography, blood cultures, and clinical manifestations [5]. The European Diagnostic and Management Guide to EI (2023) advocates for use of the modified Duke Criteria to standardize diagnosis. In our patient’s case, the diagnosis was confirmed by the presence of two major criteria: echocardiographic visualization of vegetation and persistent positive blood cultures. Notably, echocardiography exhibits a specificity of 70% in detecting native valve endocarditis [5].

Endocarditis affecting native valves predominantly afflicts men, with most patients aged over 50. Among valvopathies, mitral rheumatic valvopathy and aortic valvopathy are the most associated with endocarditis [12]. Tricuspid valve involvement in endocarditis is rare [13], making the concomitant affliction of both the tricuspid and aortic valves, as seen in our case, an unusual occurrence scarcely documented in the literature. It is postulated that the rupture of a Valsalva sinus aneurysm led to simultaneous damage to both valves during the infectious process [13].

Tricuspid valve IE comprises 5–10% of all cases of infectious endocarditis, typically linked to intravenous drug use, medical interventions, venous catheters, or hemodialysis [14]. Another at-risk group for right heart IE includes those with congenital heart disease, such as interventricular septal defects. However, none of these predisposing conditions were present in our patient. Instead, it is hypothesized that the rupture of a Valsalva aneurysm resulted in concurrent damage to both the aortic and tricuspid valves.

Complications of bacterial endocarditis encompass both cardiac and extracardiac manifestations. Cardiac complications include heart failure, myocarditis/pericarditis, perivalvular extension of infection, and conduction abnormalities (atrioventricular blocks). Extracardiac complications involve uncontrolled and persistent disease, splenic and renal embolic phenomena, and neurological complications such as cerebral vascular accidents, transient ischemic strokes, brain abscesses, meningitis, toxic encephalopathy, silent cerebral embolisms, and mycotic aneurysms [5,7]. Notably, all neurological complications, regardless of sequelae, are associated with heightened mortality [15].

Clinical manifestations encompass a spectrum of renal complications, such as glomerulonephritis, renal infarction, renal abscess, and renal insufficiency. Renal insufficiency may manifest due to hemodynamic compromise or because of antibiotic-induced toxicity, notably acute interstitial nephritis [5].

The most important manifestations of infective endocarditis (IE) include voluminous vegetations, valvular regurgitations, and systemic embolisms, all of which were observed in our patient. Additionally, the patient presented with evolving right heart failure and respiratory failure exacerbated by septic pulmonary embolisms [5].

The presence of large vegetations (>1 cm) predisposes patients to embolic events and serves as a criterion for emergent surgical intervention, particularly in the context of uncontrolled infection and severe acute congestive heart failure. Vegetations on the aortic valve can precipitate cerebral, splenic, and renal embolisms affecting peripheral, mesenteric, or coronary arteries, while vegetations on the tricuspid valve can lead to pulmonary embolisms [5,16]. Clinical presentation with altered consciousness and renal impairment suggests the occurrence of cerebral and renal embolisms alongside respiratory failure potentially exacerbated by septic pulmonary embolisms. Nephritic syndrome arises from glomerular injury mediated by the immune mechanism [5].

Successful management of IE revolves around identifying and eradicating the causative microorganism, implementing an optimal bactericidal treatment regimen, considering surgical intervention in select cases, managing adverse effects of antibiotic therapy, and addressing complications as they arise during the disease course [17].

In cases of IE complicated by heart failure, the combination of antibiotic therapy with surgical intervention enhances the likelihood of cure and survival. Cardiac surgery is also associated with improved survival in patients with large vegetations exceeding 10 mm on the mitral and aortic valves [18,19].

Indications for right heart surgery include heart failure refractory to diuretic therapy, persistent vegetation larger than 20 mm on the tricuspid valve following recurrent pulmonary embolisms, difficulty eradicating endocarditis-causing microorganisms, or persistent bacteremia lasting over seven days despite adequate antibiotic therapy.

Left heart surgery is warranted in cases of hemodynamic instability secondary to cardiac insufficiency; uncontrolled infection leading to abscess formation, pseudoaneurysm, or fistula; the presence of enlarging vegetation; infections caused by multidrug-resistant microorganisms; or persistent positive blood cultures despite treatment for prevention of embolic events in isolated, very large vegetations or large vegetations over 10 mm expressing embolic episodes despite adequate antibiotic treatment or embolic events associated with stenosis or severe regurgitation and low operative risk [16].

If there is a first infectious outbreak responsible for endocarditic infection, it should be eradicated before cardiac surgery unless emergency surgery is indicated. In our case, the patient was indicated for emergency surgery because he had large vegetation (>2 cm) at the level of the aortic and tricuspid valve [5,10]. Still, according to the patient’s clinical condition, the team of cardiovascular surgeons delayed the surgery and recommended continuing the treatment according to the antibiogram.

The prognosis of patients with IE is very variable, affected by the virulence of the etiological agent, complications, the precocity and correctness of antibiotic treatment, the option and choice of surgical treatment, and also the general status of the patient (age, comorbidities). Early diagnosis and proper treatment increase patient survival to over 70% [7,17,18].

The mortality rate of IE is 20–25%, even with adequate antibiotic treatment [19].

Predictive factors for unfavorable evolution are plurivalvular involvement, vegetation over 15 mm, hypoalbuminemia, and renal failure (creatinine over 2 mg/dL). Old age, diabetes, and symptomatic neurological complications (especially because of a stroke) are predictors of increased hospital mortality [20].

The unfavorable evolution was determined by the low immunity of the patient given by the associated pathology (liver cirrhosis, COPD), by the simultaneous involvement of both the right and left heart, by the presence of large vegetations with embolic risk, and by the pattern of antibiotic resistance of this new germ [21].

The emergence of new etiological agents requires the development of serological and molecular methods. It is necessary to improve the techniques for performing polymerase chain reactions [18].

The changed microbiological profile of IE and the emergence of strains resistant to the classical schemes proposed in the guidelines require the use of state-of-the-art microbial agents. The most common species reported to be resistant to vancomycin are nosocomial agents and methicillin-resistant strains of *Staphylococcus* and *Enterococcus* (*Enterococcus faecium* and *Enterococcus faecalis*). New antibiotics are used for the treatment of these patients. Even under these conditions, multidrug resistance to antibiotics is a real problem, with the expected solution being the production of new antibiotics active on new strains while not producing multidrug resistance in their turn [5,18].

The etiology of this IE was a vancomycin-resistant strain of *Streptococcus thoraltensis*. Due to the rare encounter of this pathogen in human infections, we performed a brief review to shed light on the existing data regarding this bacterium.

We searched PubMed, MedNar, and Cochrane Library electronic databases for publications made by 30 December 2022. We considered the following terms included in the studies’ title or abstract: “*Streptococcus thoraltensis*”, combined with the Boolean operator “AND”, along with “human infections”. We excluded studies in languages other than English, Spanish, or French. The results are summarized in Table 1.

We included nine papers that describe infections produced by *Streptococcus thoraltensis*, emphasizing the heterogenicity of the infections this pathogen may produce [22,23,24,25,26,27,28,29,30].

The first reported case of *Streptococcus thoraltensis* involvement in human infections was described by Dhotre S et al. in 2014 [22]. They isolated the bacterium from patients with periodontitis by sampling the subgingival plaque. Identification of the bacterium was performed using the Vitek 2 system, similar to our article. Analysis of the antimicrobial resistance of *Streptococcus thoraltensis* isolated from patients with periodontitis showed resistance to ampicillin, cefepime, cefotaxime, and ceftriaxone [25]. 

In 2015, Vukonich M et al. described an interesting case of *Streptococcus thoraltensis* chorioamnionitis in a 30-year-old female patient [23]. The bacterium was isolated from the placenta and from the tracheal aspirate of the newborn. Using for the first time a combination of ampicillin and gentamycin for the treatment of *Streptococcus thoraltensis* infection, they were able to provide a favorable outcome for the patient. Another particularity of this case is the evaluation of a possible risk factor. Due to its initial isolation from pigs, the authors were able to track down a possible familiar exposure [1,23]. Tracking possible risk factors is crucial in establishing the pathogenicity and epidemiological risk of new and emerging pathogens. 

Throat infections due to *Streptococcus thoraltensis* were described by Bakir et al. [24]. For bacterial identification they used conventional methods as well as the Vitek 2 system. Their work described a more extensive antimicrobial resistance profile for the isolated strains for the first time, including resistance to vancomycin [24]. This phenomenon is of particular interest considering the repercussions regarding the treatment of the infections produced by these resistant strains. However, no molecular analysis was performed to track the vancomycin resistance genes, so the epidemiologic implications of this strain remain unclear.

Identification of *Streptococcus thoraltensis* from blood samples was described for the first time by Petridis N et al. in 2018 [26]. In their article, they present the case of a 55-year-old female patient with a fever of unknown origin due to bacteriemia. Similar to the case described by Vukonich M et al., they used a combination of ampicillin and gentamycin for treatment and the evolution was favorable, with the patient eventually discharged [23,26]. This presented another possible risk factor for bacteriemia: prior traumatic dental procedures. As previously stated, analyzing risk factors is of outmost importance.

In 2019, Wazir et al. published the only article that describes the involvement of *Streptococcus thoraltensis* in a case of pneumoniae [27]. In the article, they describe the case of a 38-year-old female patient that developed symptoms of pneumoniae after giving birth. The bacterium was isolated from a blood sample and identified using the Vitek 2 system. There were no risk factors described. Regarding the treatment, initially the patient received a combination of cefepime and vancomycin. However, after the isolation and identification of the bacterium, the patient was switched to ceftriaxone with a favorable outcome. To our knowledge, this is the only reported case where the patient received only one antibiotic and had a favorable outcome [27]. 

Except for the case we presented, only two other cases of infections with *Streptococcus thoraltensis* with a fatal outcome have been described [28,29]. Hai PD et al. published an article in 2020, where they present the case of a 68-year-old male patient with infective endocarditis [28]. There are several similarities with the patient presented in our case, like the big valvular vegetations produced by the microorganism. The patient eventually succumbed to the illness despite treatment with linezolid and levofloxacin. Venkatachalam P et al. described the only case of necrotizing fasciitis due to *Streptococcus thoraltensis* [29]. The 65-year-old male patient also succumbed to the illness, though there are no data regarding the antimicrobial treatment used. 

The most recent published article involving infections produced by *Streptococcus thoraltensis* was published in 2023 by Salazar-Briseno IE et al. [30]. They presented three cases of catheter-associated urinary tract infection (CAUTI). In all the cases described, the patients were female, with age ranging between 54 and 75 years of age. All these cases were treated to a favorable outcome through a combination of antimicrobial medications involving a carbapenem (meropenem or imipenem) and another drug (levofloxacin or metronidazole). It is important to mention that Salazar-Briseno IE et al. described another case presented at a medical conference in their article, though no reference for that case was available [30]. With CAUTIs being frequent, their work described a new perspective regarding the pathogenicity *of Streptococcus thoraltensis*. This bacterium might be more frequent than previously known but has so far been disregarded as a human pathogen. In the future, a closer look should be taken when microbiology laboratories isolate a strain of *Streptococcus thoraltensis*. 

Identification of this bacterium is usually performed using the Vitek 2 system and other conventional techniques. Even if these techniques provide accurate identification, perhaps using more modern diagnostic tools (like MALDI-TOF or MALTI-TOF MS) in the future may be relevant in terms of shedding light on *Streptococcus thoraltensis* infections. Regarding the antimicrobial resistance profile of the isolated strains, the articles described resistance to a wide variety of antimicrobials. However, resistance to vancomycin was described in only two other clinical cases [24,28]. In both situations, no molecular analysis of vancomycin-resistant genes was performed. Therefore, the underlying mechanism of vancomycin resistance remains unclear in those *Streptococcus thoraltensis* isolates. 

## 4. Conclusions

*Streptococcus thoraltensis* is a rare species of the viridans streptococcus group that can cause severe infections in humans, including bacterial endocarditis. Based on a brief review, we noticed an extended variety of infections caused by *Streptococcus thoraltensis*. Concomitant damage to the aortic and tricuspid valves and large vegetation with embolic risk have been described in cases of bacterial endocarditis with this germ. Endocarditis caused by *Streptococcus thoraltensis* is challenging to treat because of the severity of the clinical form and the cause of this germ’s particular antibiotic resistance pattern. Although there are no available protocols for infections produced by this bacterium, an association of antibiotics usually provides a favorable outcome for the patient. The spotlight should focus on rare and emerging pathogens in the future to adequately tackle healthcare issues that may arise.

## Figures and Tables

**Figure 1 microorganisms-12-00566-f001:**
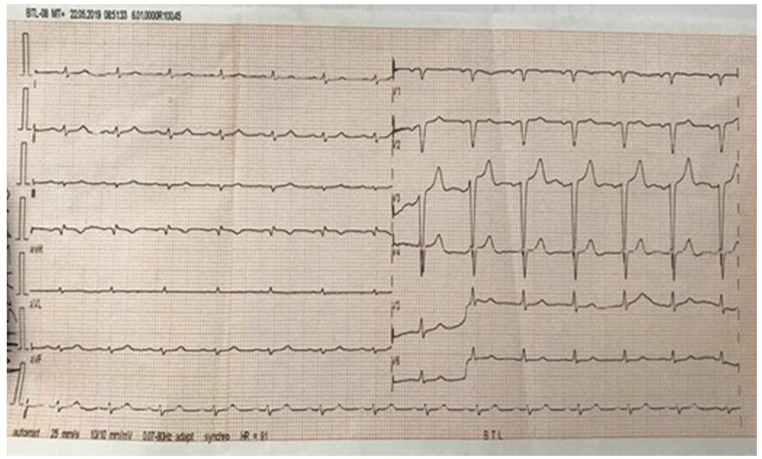
EKG: sinus rhythm, heart rate 80/min, QRS axis 90°, QS V1–V3.

**Figure 2 microorganisms-12-00566-f002:**
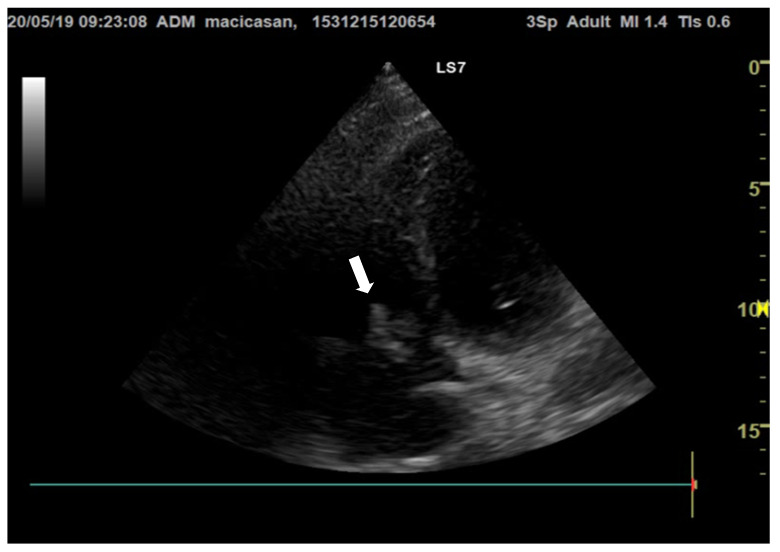
Transthoracic echocardiography, apical 4-chamber view, reveals hyperechogenic vegetations on the tricuspid valve (the white arrow is indicating the vegetation on the aortic valve).

**Figure 3 microorganisms-12-00566-f003:**
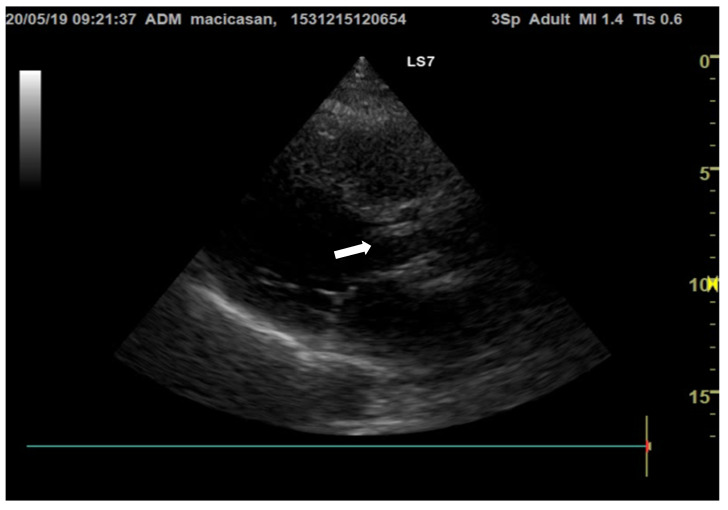
Transthoracic echocardiography, long axis parasternal section, reveals hyperechogenic vegetations on the aortic valves (the white arrow is indicating the vegetation on the aortic valve).

**Table 1 microorganisms-12-00566-t001:** Overview of *Streptococcus thoraltensis* cases in the literature.

Nr. Crt.	Author	Country	Year	Article Type	Patient Age (y.o.) Sex	Sample	Infection	Bacterial Identification	Antimicrobial Resistance	Treatment	Outcome	Possible Risk Factors
1	Dhotre S et al.	India	2014	Letter to the Editor	N/A	Subgingival plaque	Periodontitis	Vitek 2	N/A	N/A	N/A	Poor dental hygiene
2	Vukonich M et al.	United States of America	2015	Case Report	30/Female	Chorioamniotitis	Placenta, tracheal aspirate	Vitek 2	P, E	A + G	Discharged	Familial exposure to swines
3	Bakir S et al.	Iraq	2016	Original Article	N/A	Throat swab	Throat infection	Microscopy, morphology, biochemical test, Vitek 2	V, AMC, A, E, DA, P, CRO, G, IMP	N/A	N/A	N/A
4	Dhotre S et al.	India	2016	Original Article	N/A	Subgingival plaque	Periodontitis	Vitek 2	A, FEP, CTX, CRO	N/A	N/A	Poor dental hygiene
5	Petridis N et al.	Greece	2018	Case Report	55/Female	Blood	Bacteremia/fever of unkown origin	Vitek 2	P, A, CTX, CRO	A + G	Discharged	Traumatic dental procedure
6	Wazir M et al.	United States of America	2019	Case Report	38/Female	Blood	Postpartum pneumonia	N/A	N/A	FEP + V wwitched to CRO	Discharged	N/A
7	Hai PD et al.	Vietnam	2020	Case Report	68/Male	Blood	Infective endocarditis	Vitek 2	V, CIP	LZD + LEV	Deceased	N/A
8	Venkatachalam P et al.	India	2022	Case Report	65/Male	Blood	Necrotizing fasciitis	Vitek 2	TET	N/A	Deceased	N/A
9	Salazar Briseno IE et al.	Mexico	2023	Letter to the Editor	75/Female 70/Female 54/Female	Urine Urine Urine	CAUTI CAUTI CAUTI	N/A N/A N/A	N/A N/A N/A	IMP + LEV MEP + MT MEP + MT	Discharged Discharged Discharged	Urinary catheter Urinary catheter Urinary catheter
10	Present Case	Romania	2023	Case Report	65/Male	Blood	Infective endocarditis	Vitek 2	V, TET, TGC, DA, R	V + CRO wwitched to A + G	Deceased	Poor dental hygiene

Abbreviations: A, ampicillin; AMC, amoxicillin/clavulanic acid; CAUTI, catheter-associated urinary tract infection; CIP, ciprofloxacin; CRO, ceftriaxone; CTX, cefotaxime; DA, clindamycin; E, erythromycin; FEP, cefepime; G, gentamicin; IMP, imipenem; LEV, levofloxacin; LZD; linezolid; MEP, meropenem; MT; metronidazole; P, penicillin; R, rifampicin; TET, tetracycline; TGC, tigecycline; V, vancomycin.

## Data Availability

Not applicable.

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
