# Peer review of "Vancomycin-Resistant Streptococcus thoraltensis: A Case Report of Bacterial Endocarditis and Review of Literature on Infections Caused by This Pathogen"

_microorganisms, 2024, doi:10.3390/microorganisms12030566_

Round 1

Reviewer 1 Report

Comments and Suggestions for Authors

The topic proposed by the authors is interesting, since only one case of endocarditis due to S. thoraltensis had been previously described in the literature. However, most of the article deals with the characteristics of infective endocarditis and little with the review performed. Text should be trimmed in that part. Another option is to send the article as a case report.

- Line 15: viridans

- Line 25: italics

- Case report: put verb tenses in past tense

- Line 68: BMI

- Line 76: abdominal ultrasound

- Figure 1: can be deleted and describe the hallagos in the text

- Line 80: no Roman numerals

- Line 26: was

- 160: italics

- 162: italics

- 168: italics

-171,172, 172, 273, 274, 278: italics

- Table 1: the legend of the initials used must appear.

- Check bibliography, it is not uniform. It must be adapted to the standards of the journal

- The title implies that the review is about S. thoraltensis infective endocarditis, when it is about S. thoraltensis infections in humans. It could be changed to "Vancomycin-resistant Streptococcus thoraltensis infec-

tive endocarditis: a case report and brief literature review about S. thoraltensis infections".

- In the discussion, there is a description of the epidemiological, clinical and diagnostic features of infective endocarditis, which is not the aim of the article; instead the review is left in the background. I believe that this part should be trimmed as it is not relevant to the article.

- The entire nomenclature of bacterial species should be reviewed and written in italics.

Comments on the Quality of English Language

Must be revised.

Author Response

Thank you so much for reviewing our paper. We responded as follows:

The topic proposed by the authors is interesting, since only one case of endocarditis due to S. thoraltensis had been previously described in the literature. However, most of the article deals with the characteristics of infective endocarditis and little with the review performed. Text should be trimmed in that part. Another option is to send the article as a case report.

- Line 15: viridans

Thank for this observation. We corrected it.

- Line 25: italics

Thank you. We corrected it.

- Case report: put verb tenses in past tense

- Line 68: BMI

Thank you. We corrected it.

- Line 76: abdominal ultrasound

Thank you. We corrected it.

- Figure 1: can be deleted and describe the hallagos in the text

Figure 1 presents more details regarding the clinical examination of the patient, and we consider it to be mandatory in the context of a case report. The information does not repeat itself. However, we apreciate your insightful comment and decided to change the document.

- Line 80: no Roman numerals

Thank you. We corrected it.

- Line 26: was

Thank you. We corrected it.

- 160: italics

- 162: italics

- 168: italics

-171,172, 172, 273, 274, 278: italics

Thank you. We corrected all the italics needed.

- Table 1: the legend of the initials used must appear.

Thank you. We corrected it.

- Check bibliography, it is not uniform. It must be adapted to the standards of the journal

Thank you for this comment. We reviewed the bibliography accordingly.

- The title implies that the review is about S. thoraltensis infective endocarditis, when it is about S. thoraltensis infections in humans. It could be changed to "Vancomycin-resistant Streptococcus thoraltensis infective endocarditis: a case report and brief literature review about S. thoraltensis infections".

Thank you for this observation. To our knowledge there is only one additional case report of endocarditis produced by Streptococcus thoraltensis so there is not enough data to provide a review only on endocarditis. However, in the era of the emerging pathogens there is constant need to update the existing data. The title is not misleading, we formulated it like this in order to avoid unnecessary repletion. However, we decided to follow your suggestion and modify the title accordingly.

- In the discussion, there is a description of the epidemiological, clinical and diagnostic features of infective endocarditis, which is not the aim of the article; instead, the review is left in the background. I believe that this part should be trimmed as it is not relevant to the article.

Thank you for this observation. Our paper is a case report and brief review of the literature regarding the infections produced by Streptococcus thoraltensis. As mentioned before there are not enough data regarding endocarditis produced by this pathogen and we wanted to create a context for bacterial endocarditis, so the relevance of the paper becomes clear. However, we trimmed this part of discussion for a better understanding.

- The entire nomenclature of bacterial species should be reviewed and written in italics.

Thank you. We corrected it.

Reviewer 2 Report

Comments and Suggestions for Authors

 It's a case report and brief literature review. such papers have a niche audience, clinicians. The topic is about Vancomycin-resistant Streptococcus thoraltensis, a rare pathogen. The conclusions are consistent with the evidence and arguments and authors address the main question posed.

The manuscript needs some corrections as stated below, plus English editing.

There are several corrections to be made:

Line 59: Toxic liver cirrhosis, please specify.

Line 60: Long-term alcohol user, please comment.

Lines 73-75: No results available.

Line 79: biohumoral presented?

Line 115: European …., reference.

Line 136: Please specify days/blood culture repeated.

Please comment on previous hospitalization, hospital outpatient visits, and Clostridioides difficile.

Comments on the Quality of English Language

None

Author Response

Thank you for this insightful review. Here are the comments, as follows:

It's a case report and brief literature review. such papers have a niche audience, clinicians. The topic is about Vancomycin-resistant Streptococcus thoraltensis, a rare pathogen. The conclusions are consistent with the evidence and arguments and authors address the main question posed.

The manuscript needs some corrections as stated below, plus English editing.

There are several corrections to be made:

Line 59: Toxic liver cirrhosis, please specify.

Thank you. We added chronic alcoholism as the main cause for toxic liver cirrhosis.

Line 60: Long-term alcohol user, please comment.

            Long-term alcohol use  represent Long-Term Health Risks. Over time, excessive alcohol use can lead to the development of chronic diseases and other serious problems including: high blood pressure, heart disease, stroke, liver disease. We have mentioned: “the patient had a history of long-term alcohol consumption, with daily intake exceeding 40 grams for the past 40 years”.

Lines 73-75: No results available.

We added:

The biohumoral analyses revealed several significant findings:

  • Presence of an inflammatory syndrome, evidenced by elevated levels of C-reactive protein (CRP) at 9.72 mg/dL, an erythrocyte sedimentation rate (ESR) of 27 mm/h, a leukocytosis of 10.23 x 10^9/L with 90.7% neutrophils (within the range of 30-75%), and an elevated ferritin level of 627 ng/mL.
  • Normochromic normocytic anemia, indicated by a hemoglobin (Hb) level of 10.6 g/dL.
  • Thrombocytopenia, characterized by a platelet count of 35 x 10^9/L.
  • Evidence of hepatic impairment, including hypoalbuminemia with an albumin level of 2.61 g/dL, decreased cholinesterase activity at 1349 IU/L, and an elevated international normalized ratio (INR) of 1.56.
  • Intrahepatic cytolysis, demonstrated by elevated levels of both direct bilirubin (Bd) at 2.72 mg/dL and total bilirubin (Bt) at 5.3 mg/dL.
  • Renal dysfunction classified as stage 3b according to KDIGO criteria, with elevated serum creatinine levels of 2.22 mg/dL, a calculated creatinine clearance of 28 mL/min, hyperkalemia with a potassium (K) concentration of 5.8 mmol/L, hyponatremia with a sodium (Na) concentration of 126 mEq/L, and hyperuricemia with a uric acid level of 18.10 mg/dL.
  • Elevated levels of N-terminal pro-B-type natriuretic peptide (NT-proBNP) at 1747 pg/dL, suggesting cardiac dysfunction.
  • Abnormal urine analysis revealing leukocyturia, hematuria, and significant proteinuria at 2g/24h.
  • Serological testing indicating positivity for hepatitis B surface antigen (HBsAg), negativity for anti-hepatitis C virus (HCV) antibodies, and negativity for human immunodeficiency virus (HIV) antibodi
  • GDH and toxin A, B tests for Clostridium difficile were negative.

Line 79: biohumoral presented?

Thank you. We corrected it to biohumoral analysis.

Line 115: European …., reference.

Thank you. We corrected it - Ww added reference (5).

Line 136: Please specify days/blood culture repeated.

At the presented line there are no data regarding the blood cultures. In line 99 we mentioned  *All three sets of blood cultures were collected at an interval of one hour*

Please comment on previous hospitalization, hospital outpatient visits, and Clostridioides difficile.

- GDH and toxin A, B tests for Clostridium difficile were negative - we added

- A syncopal episode occurred six weeks before the ED presentation. The patient was hospitalized, and following ECG holter monitoring, the diagnosis of repetitive paroxysmal FIA was established. Echocardiography revealed the appearance of dilated cardiomyopathy without visualizing transthoracic valvular vegetation at the level of valves. At the same time, the patient was diagnosed with toxic liver cirrhosis Child A and BOPC stage II GOLD. The above treatment has been initiated.

The patient did not come for an outpatient check-up.

Round 2

Reviewer 1 Report

Comments and Suggestions for Authors

The authors have responded correctly to the comments and suggestions made. It would only remain to change some italics and to clarify, in the introduction, which paragraph would definitely remain in the manuscript and which would be eliminated (lines 49-62 or lines 63-81), since sentences and concepts are repeated.

After that, it could be accepted for publication

295: Staphylococcus aureus in italics

302: Streptococcus viridans in italics

476: italics

Author Response

Thank you for the kind words.

We revised the introduction and used italics where they were missing.